# Which Agent Causes Task Failures and When?
# On Automated Failure Attribution of LLM Multi-Agent Systems

**Shaokun Zhang** [* † 1]   **Ming Yin** [* 2]   **Jieyu Zhang** [3]   **Jiale Liu** [1]   **Zhiguang Han** [4]   **Jingyang Zhang** [2]   **Beibin Li** [5]
**Chi Wang** [6]   **Huazheng Wang** [7 8]   **Yiran Chen** [2]   **Qingyun Wu** [1 8]

## Abstract

Failure attribution in LLM multi-agent systems—identifying the agent and step responsible for task failures—provides crucial clues for systems debugging but remains underexplored and labor-intensive. In this paper, we propose and formulate a new research area: **automated failure attribution** for LLM multi-agent systems. To support this initiative, we introduce the **Who&When** dataset, comprising extensive failure logs from 127 LLM multi-agent systems with fine-grained annotations linking failures to specific agents and decisive error steps. Using the Who&When, we develop and evaluate three automated failure attribution methods, summarizing their corresponding pros and cons. The best method achieves 53.5% accuracy in identifying failure-responsible agents but only 14.2% in pinpointing failure steps, with some methods performing below random. Even SOTA reasoning models, such as OpenAI o1 and DeepSeek R1, fail to achieve practical usability. These results highlight the task's complexity and the need for further research in this area. Code and dataset are available in the public repository.

## 1. Introduction

In recent years, the reframing Large Language Models (LLMs) as agents and built agentic multi-agent systems—composed of interactive, LLM-powered agents collaborating to achieve shared goals—has garnered significant attention (Hong et al., 2023; Li et al., 2023a; Wu et al., 2023). These purposefully designed agentic systems have demon-

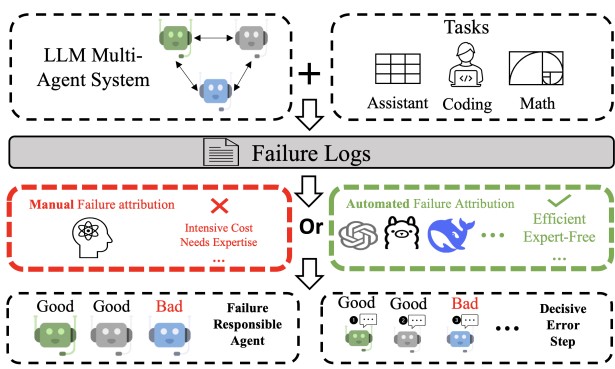

*Figure 1.* When developing LLMs-powered multi-agent systems, failure attribution—identifying system components responsible for task failures based on evaluation results—has received limited attention in existing research. This process is typically performed manually, demanding substantial labor and specialized expertise. In this study, we explore the potential for automating this process.

strated remarkable potential across various domains, including coding (Hong et al., 2023), scientific discovery (Ghafarollahi & Buehler, 2024), and addressing complex real-world challenges (Fourney et al., 2024).

Once constructed, these systems are typically refined through an iterative process when they fail in specific scenarios: evaluation against established benchmarks, followed by manual failure attribution and system refinement. This cycle repeats until the desired outcomes are achieved. Failure attribution—identifying the components of the system that directly lead to task failures—is a crucial step that serves as the foundation for guiding improvements. Despite its importance, this process is largely overlooked in mainstream research, which typically leaves it as a manual task requiring significant labor, such as analyzing complex historical logs and navigating the technical intricacies of the system. Moreover, mapping benchmark evaluation results to failure components is heavily dependent on domain expertise, imposing additional requirements on practitioners. As systems grow in complexity, this challenge becomes increasingly difficult due to the growing number of components that must be considered when conducting failure attribution.

Previous manual efforts involve a non-straightforward way

---

[*]Equal contribution  [1]Pennsylvania State University [2]Duke University [3]University of Washington [4]Nanyang Technological University [5]Meta [6]Google DeepMind [7]Oregon State University [8]AG2AI, Inc.. Correspondence to: [†]Shaokun Zhang <shaokun.zhang@psu.edu>.

*Proceedings of the 42nd International Conference on Machine Learning*, Vancouver, Canada. PMLR 267, 2025. Copyright 2025 by the author(s).

to facilitate failure attribution in multi-agent systems: developing increasingly fine-grained benchmarks, with the hope that more metrics will enable quicker failure attribution (Zhuge et al., 2024). For example, DevAI (Zhuge et al., 2024) introduces a coding benchmark that incorporates diverse delivery requirements, offering a more nuanced evaluation compared to the widely used SWE-Bench (Jimenez et al., 2023), which focuses exclusively on final resolution rates. Despite these advancements, the process of failure attribution based on benchmark results remains a manual process, merely providing more metrics as reference points without fundamentally addressing the underlying challenges. With increasingly comprehensive benchmarks, a fundamental question remains unanswered: **which components of the agentic system require improvement?**

We argue that evaluation and failure attribution should be tightly integrated, adhering to the principle that "evaluation is not an end in itself, but a means to improvement." (Scriven, 1991) More research efforts should focus on bridging the substantial gap between evaluation results and failure attribution, which currently relies heavily on manual labor. Drawing inspiration from the LLM-as-a-judge paradigm (Gu et al., 2024; Zheng et al., 2023), which leverages LLMs to replace human effort in evaluation, we propose to bridge the gap between evaluation and failure attribution by harnessing the comprehensive judgment capabilities of LLMs. Specifically, we propose and formulate a new research problem: **automated failure attributions** in LLM multi-agent systems. When failures occur under specific scenarios during evaluation, the goal is to automatically identify the components responsible for these failures without human intervention. We believe this research could serve as a substitute for manual failure attribution, enabling human resources to focus on improving system functionality rather than performing time-intensive diagnostics as shown in Figure 1

To advance the research in this area, we introduce the **Who&When** benchmark, comprising extensive failure logs annotated with fine-grained failure details for addressing real-world tasks, where these logs are collected from 127 LLMs-powered multi-agent systems. This benchmark includes both algorithmically generated and hand-crafted multi-agent systems, encompassing a wide range of realistic scenarios. Each failure log is accompanied by meticulous annotations, specifying the failure-responsible agent, the corresponding failure step, and the reasons for failure in plain language. The primary task involves pinpointing the agent most accountable for the failure and the exact step where the error occurred. Predicting the failure-responsible agent serves as a fundamental requirement for failure attribution, given that agents are the basic units of multi-agent systems. Extending this to the specific failure step prediction imposes a higher level of requirement, enabling more fine-grained failure attribution, such as uncovering the specific reasons behind failures, which can further facilitate targeted system refinements. We believe that Who&When can serve as a foundational resource for driving progress in automated failure attribution research.

Additionally, we construct and evaluate several automated failure attribution approaches on the Who&When. Our findings reveal the strengths and limitations of each method, as well as their performance across different conditions, including model variations, historical context lengths, and the presence or absence of query labels. The results underscore the complexity of using LLMs for failure analysis in multi-agent systems. For example, the best-performing method achieved only 8.77% accuracy in identifying decisive error steps within the hand-crafted agentic system.

## 2. Problem Formulation: Automated Failure Attribution in Multi-Agent Cooperation

In this section, we introduce decisive errors and formulate the automated failure attribution problem. We adopt the widely-adopted turn-based LLM multi-agent protocol (Hong et al., 2023; Li et al., 2023a; Wu et al., 2023).

**Background.** Considering a LLMs-powered multi-agent system $\mathcal{M}$ with a group of $N$ agents, denoted as $\mathcal{N} = \{1, 2, ..N\}$, that operate at discrete time steps. These agents are taking actions in a turn-based protocol, meaning that exactly one agent performs an action at each time step. Formally, the system is described as:

$$\mathcal{M} = \left\langle \mathcal{N}, S, A, P, \phi \right\rangle. \tag{1}$$

Here, $S$ is the set of possible states. $A$ is the global action set; each agent $i \in \mathcal{N}$ can typically perform actions from some subset $A_i \subseteq A$. $\phi(t)$ is a function that indicates which agent is active at time $t$, thus specifying the turn-based rule. $P\big(s_{t+1} \mid s_t, a_t, \phi(t)\big)$ is the state-transition probability, given that *only one* agent $\phi(t)$ acts at time $t$.

We employ $\phi(t)$ to denote the agent that takes an action $a_t$ at time step $t$. A full trajectory $\tau$ can be written as: $\tau = \big(s_0, a_0, s_1, a_1, \ldots, s_T\big)$, where $T$ is a terminal time step or when the system enters a terminating state.

**Decisive Error and Objective.** We use a tuple $(i, t)$ to denote a mistake in a trajectory, which means agent $i$ is active at time $t$, and its action $a_t$ is deemed a mistake (e.g., wrong reasoning etc.). A trajectory may contain multiple mistakes, but not all of them result in overall failure. We employ $Z(\tau)$ to denote the result of a trajectory $\tau$.

$$Z(\tau) = \begin{cases} 1, & \text{if the system ultimately fails,} \\ 0, & \text{otherwise.} \end{cases} \tag{2}$$

Suppose the original trajectory $\tau$ is a failure, i.e., $Z(\tau) = 1$. Considering the following scenario, if correcting the mistake made by agent $i$ at time $t$: we replace $a_t$ with a "correct" action $\tilde{a}_t$. The steps prior to step $t$ remain unchanged, while the actions following $t$ are adjusted accordingly to ensure correctness. This process generates a modified trajectory:

$$\tau^{(i,t)} = \mathcal{I}_{(i,t)}(\tau), \tag{3}$$

where $\mathcal{I}_{(i,t)}$ denotes the intervention. If in the modified trajectory we obtain $Z\big(\tau^{(i,t)}\big) = 0$ (success), then the error $(i, t)$ is said to be a *decisive error*. Formally, we define the *decisive error indicator* $\Delta_{i,t}(\tau)$ as

$$\Delta_{i,t}(\tau) = \begin{cases} 1, & \text{if } Z(\tau) = 1 \text{ and } Z\big(\tau^{(i,t)}\big) = 0, \\ 0, & \text{otherwise.} \end{cases} \tag{4}$$

In words, $\Delta_{i,t}(\tau) = 1 \iff$ Fixing agent $i$'s mistake at time $t$ changes $Z(\tau)$ from 1 (fail) to 0 (success). Formally, the decisive error could be defined as agent-time pairs $(i^*, t^*)$ such that $\Delta_{i^*,t^*}(\tau) = 1$, where $i^*$ represents the agent responsible for the system failure, and $t^*$ represents the exact time step at which the critical mistake occurs. We refer to these as the *failure-responsible agent* and the *decisive error step*, respectively across the paper.

In practice, multiple decisive errors may occur within a trajectory. In our study, we address this situation by identifying the earliest error in time as the principal cause of failure. Specifically, we define an objective to determine:

$$\mathcal{C}(\tau) = \big\{ (i,t) \mid \Delta_{i,t}(\tau) = 1 \big\}, \tag{5}$$

$$(i^*, t^*) = \arg \min_{(i,t) \in \mathcal{C}(\tau)} t.$$

which selects the pair $(i^*, t^*)$ yielding the highest decisive error indicator with earliest time step. In this study, the research problem focuses on the automatic identification of the $(i^*, t^*)$ in LLMs-powered multiple agent systems.

## 3. The Who&When Dataset

To advance research in this area, we introduce a dataset called Who&When. This dataset comprises extensive failure logs from 127 LLM multi-agent systems including both algorithm-generated and human-crafted systems. These logs are carefully annotated with labels that identify the failure-reponsible agents and the decisive error steps in agent co-operation directly responsible for problem-solving failures. Additionally, each annotation is supplemented with natural language explanations, culminating in 184 distinct failure annotation tasks. The dataset is specifically designed to detect the failure-reponsible agents (who) and the corresponding steps (when) within each failure log.

Specifically, each instance in Who&When includes the following entry: **(1) Query:** A query from GAIA (Mialon et al., 2023) or AssistantBench (Yoran et al., 2024), describing a real-world question. **(2) Failure log:** The full conversation log of a specific system as it fails to solve the query. **(3) Agentic system information:** For algorithm-generated systems, including system prompts, tools, and agent names, all tailored to this specific query. **(4) Annotations:** An annotation of the agent responsible for task failure, specifying the step where the failure occurred, along with a plain-language explanation of why the failure took place. An example of the instance in this benchmark could be found in Appendix C.

To better reflect our definition of decisive error in Section 2, we design three metrics to evaluate the performance of various failure attribution methods: **(1) Agent-Level Accuracy:** This metric measures the percentage of correctly predicted failure-responsible agents by failure attribution algorithms. **(2) Step-Level Accuracy:** This metric quantifies the percentage of correctly identified decisive error steps. It imposes higher requirements on the algorithms compared to the first metric. **(3) Step-Level Accuracy with Tolerance:** To account for slight deviations, this metric allows a tolerance range for mistake step predictions. If the predicted step falls within the specified tolerance range of the actual mistake step, the prediction is considered correct.

### 3.1. Agentic Systems Constructions

Who&When includes two types of agentic systems: algorithm-generated agentic systems and one meticulously hand-crafted agentic systems, totaling 127 agentic systems equipped with diverse tools for evaluation.

**Algorithm-Generated Agentic Systems.** To ensure an adequate number of agentic systems for the Who&When datasets, we first employ the CaptainAgent algorithm (Song et al., 2024) from the AG2 library [1] to automatically generate agentic systems for each data instance sourced from the validation sets of the GAIA (Mialon et al., 2023) and AssistantBench (Yoran et al., 2024) benchmarks. Specifically, it constructs a team of agents tailored to a given task, assigning appropriate agent names, prompts, and necessary tools. The system iteratively optimizes the agents' configuration until the task is successfully completed. In the Who&When, we select only the final multi-agent configurations, along with the corresponding execution history, as these represent the optimized solutions for each query. All agents within the constructed systems, as well as the CaptainAgent algorithm itself, are based on the `GPT-4o` on `2024-08-01-preview` version. Additionally, since the primary objective of the Who&When is to capture mistakes made by agents that lead to failures in solving real-

---

[1] https://github.com/ag2ai/ag2

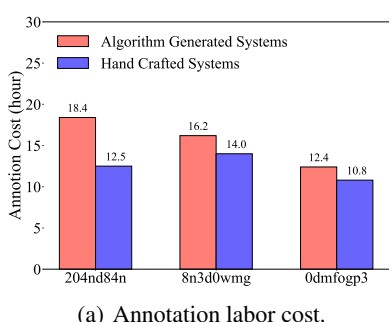

(a) Annotation labor cost.

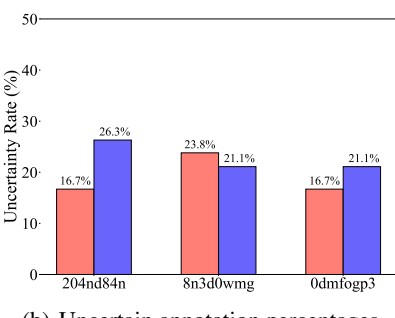

(b) Uncertain annotation percentages.

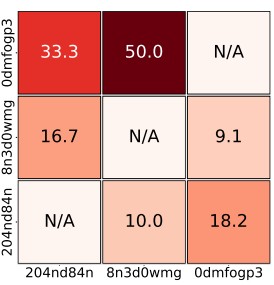

(c) Disagreement rates in voting.

*Figure 2.* Statistical analysis of the annotation process: **(1)** Total labor cost for annotations in human hours. **(2)** The proportion of uncertain annotations to total annotations during the second round. **(3)** Initial disagreement rates between annotators (note that we make sure to reach a consensus through a careful discussion and voting process afterwards). These results highlight the challenges involved in performing manual failure attribution.

world problems within agentic systems, we retain only those agentic systems that fail to successfully address the queries associated with each data instance from these benchmarks.

**Hand-Crafted Agentic Systems.** In addition to algorithm-generated systems, Who&When also includes a meticulously hand-crafted, mature multi-agent system, Magnetic-One (Fourney et al., 2024), to ensure the representation of realistic and highly refined agentic systems. Magnetic-One is a generalist agentic system designed to handle a broad range of tasks. It comprises five carefully crafted agents, each specializing in distinct capabilities, such as operating a web browser or navigating local files. We evaluate Magnetic-One using the validation set from the Assistant-Bench (Yoran et al., 2024) benchmark, aggregating its failure logs for subsequent annotation. We also test Magnetic-One on a randomly sampled subset of 30 instances from the GAIA (Mialon et al., 2023), incorporating the corresponding execution failure logs into the dataset. We exclude the rest of the GAIA dataset due to the complexity of annotating the long context logs produced by Magentic-One.

### 3.2. Decisive Error Annotation

After obtaining the failure logs of various agentic systems, we introduce an annotation procedure to identify the decisive error failure and decisive error step. To ensure precise annotation, we conduct multiple rounds of annotation performed by three human experts in AI agent (whose identities are anonymized as 0dmfogp3, 8n3d0wmg, and 204nd84n).

**Round I:** In the first round, we distribute the failure logs from all agentic systems for each query equally among three experts. To ensure consistency, we provide the experts with a standardized annotation guideline as shown in Appendix F. Each expert is tasked with annotating three elements: the single erroneous agent primarily responsible for the task failure, the specific step at which the error occurred, and the

reasoning behind the mistake in natural language. Additionally, each expert is required to categorize their annotations into two groups: those they are undoubtedly confident are correct and those they have any uncertainty about. **Round II:** In the second round, people are instructed to make an agreement on all the uncertain annotations from Round II. For these uncertain annotations, we engage in a collaborative discussion to reach a consensus. We do not simply follow the principle of majority rule; instead, we aim to ensure that everyone is persuaded and that a consensus is ultimately reached. **Round III:** In the final round, a cross-validation procedure is employed. Each expert is asked to go through another expert's annotations to assess the consistency of the annotation standards. If any discrepancies or issues with the annotations are identified, the experts engage in further discussion and, if necessary, re-annotate the data according to the established guidelines until a consensus is reached. Incorporating the viewpoints of multiple annotators and ensuring consensus among them, we aim to accurately reach the actual ground truth, as suggested by previous studies (Clemen, 1989; Zhuge et al., 2024).

### 3.3. Analysis

Annotating the decisive error agent and identifying the specific step of the error is challenging for both non-expert people and domain experts. The annotators must parse complex logs, follow the problem-solving logic of each agent, and assess whether each action is correct or if it misleads the entire problem-solving process. For example, if an agent uses a web browser to gather essential information for problem-solving, annotators must check the browser history and visit each website to determine whether the failure is due to unavailable information on the website or because the agent failed to retrieve it. As shown in Figure 2(a), three annotators spent 30.9, 30.2, and 23.2 human hours, respectively, to complete the annotations. This demonstrates that

the annotation process is very time-consuming, leading us to consider doing research on automated failure attribution.

Additionally, in many data instances, it's not just one agent that makes mistakes, but several agents. People need to identify these mistakes and select the most severe ones, which can directly lead to problem-solving failures as formulated in Section 2. Since the severity of mistakes may be subtle and even subjective at times, the process becomes even more difficult. As shown in Figure 2(b), we present the uncertain annotation percentages for three individuals. The uncertain percentages across different annotators range from 15% to 30%. We also visualize the disagreement rates between different individuals when voting on each other's uncertain data in Figure 2(c). We can see some disagreement remains before discussing to make the agreement, further highlighting the difficulties involved in the annotation process.

# 4. Can LLMs help identify *When and Which* agent causes task failures?

As revealed in Section 3.3, detecting the failure-responsible agent and corresponding failure step in agentic system are often subtle, requiring significant human effort. Given these challenges, we were thinking of performing automated failure attribution, using LLMs themselves to detect these errors and provide signal for human to perform essential improvement. In this section, we set up experiments to answer a fundamental question: *Can LLMs help identify* when and which *agent causes task failures in multi-agent systems?*

## 4.1. LLMs for Failure Attribution in Agentic Systems

To answer the question mentioned above, we propose three judgement methods for automated failure attribution in agentic systems. Through extensive experiments, we demonstrate that each method has distinct advantages and limitations, and they can be applied either independently or in combination. Furthermore, we analyze the performance of these methods across various scenarios and constraints, highlighting their applicability in different contexts.

**(1) All-at-once:** An LLM is provided with a query and the complete failure log, and it is tasked with identifying the failure-responsible agent as well as the specific step where the decisive error occurred. **(2) Step-by-step:** An LLM is provided with a query, and the failure log is presented step-by-step. At each step, the LLM is tasked with determining whether a mistake has occurred in the current step. If a mistake is identified, the judging process terminates, and the responsible agent's name along with the current step number are returned as the output. Otherwise, the process continues until the final step is reached. **(3) Binary search:** Alternatively, this method uses a receptive field approach that lies between the previous two methods. Starting with

the query and the full failure log, the LLM is tasked with determining whether the mistake occurred in the upper half or lower half of the failure logs. Once this decision is made, the LLM is provided with the selected half of the log and the process is repeated iteratively until a single step is identified. The three algorithms and corresponding prompts are detailed in Appendix A and Appendix G.

## 4.2. Settings

**Scenario.** We conduct experiments under two distinct settings to simulate various realistic scenarios. **(1) With Ground Truth:** In this setting, the final ground truth of the query that the agentic system is attempting to resolve is available to the LLMs. Our focus here is on the typical AI system development cycle, where it is common practice to use a development dataset with ground truth to identify and debug potential errors in experimental systems. **(2) Without Ground Truth:** In the second setting, the final ground truth of the query is unavailable. In this scenario, LLMs are employed to perform failure attribution in agentic systems based on their running logs. This capability can also be viewed as a form of self-reflection (Huang et al., 2023; Shinn et al., 2024), which contributes to the improvement of multi-agent systems. Throughout this paper, unless otherwise specified, all results are reported as the average accuracy across these two scenarios.

**Models.** The primary experiments are conducted using the `GPT-4o` model, unless otherwise specified. Additionally, we also incorporate several other models, including both open-source (such as the `Llama` and `Qwen` series) and closed-source models (`GPT` series), to ensure the consistency of the conclusions drawn from the experiments. Additionally, we employ advanced reasoning models, i.e., OpenAI o1 and DeepSeek R1, to assess the performance of reasoning models on failure attribution tasks. The results of these evaluations are provided in Appendix 4.8.

## 4.3. Overall Performance

We first perform experiments to compare the performance of three failure attribution methods on Who&When dataset with `GPT-4o` model. The results are reported on Table 1.

**Agent-Level Accuracy Relies on Large Receptive Field.** As shown in Table 1, all-at-once significantly outperforms the other two failure attribution methods in agent-level accuracy. Specifically, its agent-level accuracy is 19.13% and 20.69% higher than step-by-step when judging with ground truth, and 25.1% and 20.69% higher when judging without ground truth, respectively. The performance of the binary search method falls between these two approaches.

These results can be attributed to the fact that predicting the

| Agentic Systems Types | With Ground Truth | | Without Ground Truth | |
|---|---|---|---|---|
| | Algorithm Generated | Hand Crafted | Algorithm Generated | Hand Crafted |
| Random | | | | |
| Agent-Level Accuracy | 29.10 | 12.00 | 29.10 | 12.00 |
| Step-Level Accuracy | 19.06 | 4.16 | 19.06 | 4.16 |
| All-at-Once | | | | |
| Agent-Level Accuracy | **54.33** | **55.17** | **51.12** | **53.44** |
| Step-Level Accuracy | 12.50 | 5.26 | 13.53 | 3.51 |
| Step-by-Step | | | | |
| Agent-Level Accuracy | 35.20 | 34.48 | 26.02 | 32.75 |
| Step-Level Accuracy | **25.51** | **7.02** | 15.31 | **8.77** |
| Binary Search | | | | |
| Agent-Level Accuracy | 44.13 | 51.72 | 30.11 | 36.21 |
| Step-Level Accuracy | 23.98 | 6.90 | **16.59** | 6.90 |

*Table 1.* Performance of the three failure attribution methods on the Who&When dataset with and without labels, evaluated on the `GPT-4o` model. For agent-level accuracy, all-at-once outperforms binary search, which in turn surpasses step-by-step. Conversely, for step-level accuracy, step-by-step achieves the best performance, followed by binary search and then all-at-once.

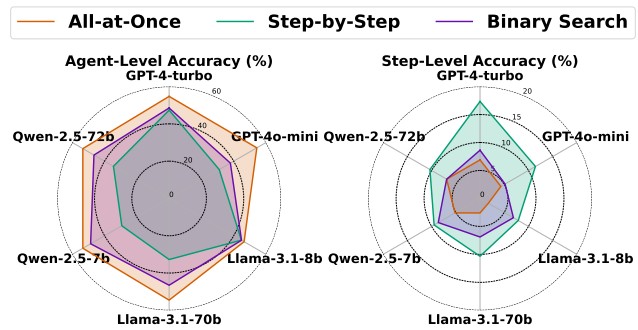

*Figure 3.* Performance comparison of three failure attribution methods on different models in both two metrics. We found the conclusion is mostly consistent with Table 1.

failure-responsible agent requires the judge LLMs to consider a broader context, including the behaviors of multiple agents. Since all-at-once has access to the entire conversation log when making the final judgment, its prediction of the failure-responsible agent is more accurate. In contrast, the step-by-step method processes the conversation history incrementally, while the final decision can be made with incomplete information, thus resulting in lower performance. Moreover, all failure attribution methods outperform the random baseline, highlighting that these approaches are nontrivial and affirming the necessity of involving LLMs for failure attribution.

> *Finding 1:* Providing broader failure log context enables more accurate agent-level failure attribution by incorporating more complete information.

**Fine-Grained Predictions Boost Better Step-Level Accuracy.** In terms of step-level accuracy, the all-at-once approach performs obviously the worst, even with its average performance falling below that of random method. This outcome underscores the method's practical infeasibility. By contrast, the step-by-step approach achieves the highest performance, outperforming other methods in step-level accuracy in 3 out of 4 cases. The performance of the binary search method still falls between these two approaches.

These two results can be attributed to the 'space-in-the-needle' problem, where LLMs often struggle to retrieve specific information from long contexts (Nelson et al., 2024). The all-at-once method has access to the largest context when making decisions about the decisive error step, but this can lead to difficulties in pinpointing the exact failure step within a long history. In contrast, the step-by-step method processes the context incrementally, allowing for more focused decision-making. The binary search method performs at a level between these two approaches.

> *Finding 2:* Incrementally processing context enables better step-level failure attribution since LLMs struggle to retrieve information from long contexts.

**Impact of Ground Truth on Failure Attribution.** We also observed that failure attribution accuracy is higher for all three methods when ground truth are available, compared to when judgments are made without ground truth in all cases in all metrics. Although the answers to users' queries may not serve as definitive 'golden labels' for each agent's correct behavior, they provide a useful reference signal for the judgment LLMs. For instance, if an agent leads the system in a completely wrong direction, with no possibility of reaching the correct final answer, the label information can directly help alert the judgment LLMs to this error.

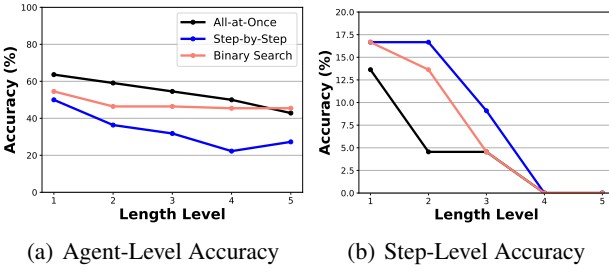

(a) Agent-Level Accuracy    (b) Step-Level Accuracy

*Figure 4.* Comparison of three failure attribution methods applied to all failure logs from the hand-crafted systems in the Who&When, evaluated under varying failure log lengths across both metrics.

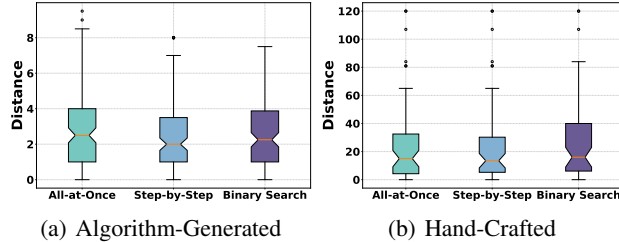

(a) Algorithm-Generated    (b) Hand-Crafted

*Figure 5.* The distances between human-annotated decisive error steps and the predicted steps for each date instance on failure logs from both algorithm-generated and hand-crafted systems.

Without such intervention, the entire system might proceed in the wrong direction without any external warning.

**Consistency of Conclusions Across Various LLMs.** In addition to the `GPT-4o` model, we conducted evaluations on other LLMs, including open-source models (e.g., the Llama series and Qwen series) as well as closed-source models (e.g., the GPT series). Due to the significant computational and token costs, we only perform experiments on hand-crafted agentic systems from Who&When which has fewer failure logs. The results of three methods are shown in Figure 3. We found that the phenomena observed in Table 1 hold consistently across different LLMs. Specifically, for agent-level accuracy, the ranking is: all-at-once, followed by binary search, and then step-by-step. Conversely, for step-level accuracy, the ranking is: step-by-step, followed by binary search, and then all-at-once.

> **Finding 3:** The pros and cons of different failure attributions methods in this study are mostly consistent across different LLMs.

### 4.4. Performance Across Varying Context Lengths

We investigate the relationship between the length of failure logs and the corresponding failure attribution performance. Specifically, the failure logs of hand-crafted agentic systems from the Who&When dataset are divided into five levels, with context length progressively increasing from Level 1 to Level 5. Specifically, Level 1 spans 5–17 steps, Level 2 covers 19–29, Level 3 includes 31–49, Level 4 ranges from 51–91, and Level 5 spans 93–130 steps. Both agent-level and step-level judgment performances across the three evaluation methods are presented in Figure 4. Algorithm-generated systems are excluded from this analysis due to their limited maximum step count of 10, which prevents meaningful divisions of context length.

Our findings indicate that all three methods exhibit a decline in both metrics as context length increases. Notably, step-level accuracy is more sensitive to context length changes

than agent-level accuracy. Furthermore, the step-by-step performance decline is particularly pronounced compared to the other two. We also analyze the distances between human-annotated decisive error steps and the predicted steps for each data instance, as shown in Figure 5. These results demonstrate that the step-by-step method outperforms the other two methods in accurately predicting the decisive error steps. However, as context length reaches its maximum, all three failure attribution methods converge to near 0%, as shown in Figure 4.

> **Finding 4:** Failure attribution performance declines as context length increases, with step-level accuracy being more sensitive.

### 4.5. Step-Level Accuracy Under Different Tolerances

| Toler. | All-at-Once | Step-by-Step | Binary Search |
|---|---|---|---|
| ± 1 | 12.07 | **14.66** | 13.79 |
| ± 2 | **19.83** | 16.38 | 18.97 |
| ± 3 | **30.17** | 18.10 | 22.41 |
| ± 4 | **37.07** | 31.90 | 31.89 |
| ± 5 | **43.10** | 33.62 | 36.21 |

*Table 2.* Step-level accuracy with different tolerances on the failure logs of hand-crafted agentic systems from Who&When dataset.

In practice, directly identifying the exact decisive error step is not always necessary; it is often sufficient to determine a range of steps where the mistake might occur. In this section, we show the performance of the three failure attribution methods under varying tolerance conditions on the failure logs of hand-crafted agentic systems from the Who&When dataset. Algorithm-generated systems are excluded from this analysis because their maximum step count is limited to 10, and increasing the tolerance would lead to artificially inflated accuracy.

As shown in Table 2, our findings show that step-by-step achieves the highest performance when the tolerance is set to 0 or 1. However, as the tolerance increases, the advantages of all-at-once become more pronounced, while the benefits

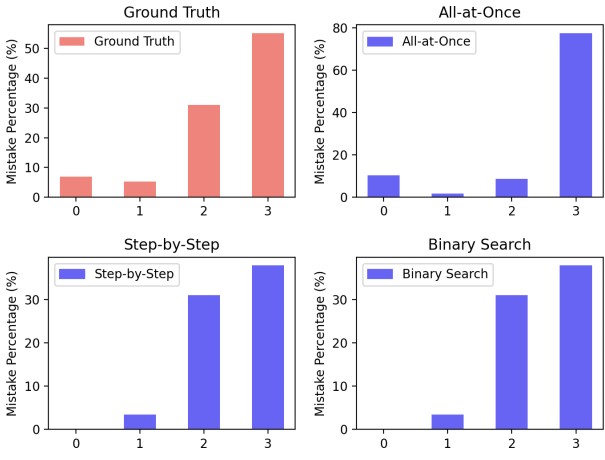

*Figure 6.* Histogram of the actual and predicted failure-responsible agents for all three methods. We present only the failure logs of hand-crafted systems in Who&When to aggregate the largest number of results for one multi-agent system. Number 0, 1, 2, 3 represents `Assistant`, `FileSurfer`, `Orchestrator` and `WebSurfer` respectively.

of step-by-step diminish. Compared to all-at-once, step-by-step demonstrates better alignment with accurate predictions when high precision is required.

> **Finding 5:** Allowing tolerance in failure attribution enables broader context processing methods to achieve competitive step-level accuracy.

### 4.6. A Statistical Viewpoint on Failure Attribution

This study primarily perform experiments on single-data-level failure attribution in LLM-powered multi-agent systems, i.e., identifying the specific component (referred to as the failure-responsible agent) and the precise location (the decisive error step) responsible for task failure in a single data instance. This practice indeed mirrors human procedures for failure attribution and could serves as a foundational tool for deriving statistical-level conclusions. Therefore, we think of whether these methods could be applied to entire datasets to extract meaningful statistical results.

In Figure 6, we show the histogram of actual and the predicted failure-responsible agents for all three methods. We only show the failure logs of hand-crafted systems from Who&When to aggregate the largest number of results for one system type. We observe that the single agent that make the most decisive errors predicted by all methods to are consistent with the ground truth (agent 3). Moreover, the top two failure-responsible agents predicted by three methods are also consistent with the ground truth in most cases (2 out of 3). These experiments demonstrate that, although the instance-level failure attribution results are not highly

positive, all three methods still yield meaningful insights from a statistical perspective. In practice, these statistical results provide a more actionable basis for system refinement compared to focusing solely on single data instances.

> **Finding 6:** The three baseline methods are more effective at performing failure attribution at a statistical level than at an instance level.

### 4.7. Can We Combine Multiple Failure Attribution Methods?

| Metrics | Cost Token Num | Agent-Level Accuracy | Step-Level Accuracy |
|---|---|---|---|
| Binary Search | 34,659 | 43.97 | 6.90 |
| △ All-at-Once | **17,106** | **57.02** | 4.39 |
| □ Step-by-Step | 87,720 | 35.96 | 7.90 |
| Hybrid Method (□&△) | 149,177 | **57.02** | **12.28** |

*Table 3.* Comparison of the three failure attribution methods with a hybrid approach that combines all-at-once and step-by-step on the failure logs of hand-crafted systems from the Who&When dataset. The hybrid method achieves the highest performance in both two metrics but incurs the highest token costs.

We then investigate whether a hybrid method could leverage the advantages of both two different methods, all-at-once and step-by-step. The former excels at failure-responsible agent predictions, while the latter is better at accurately predicting the decisive error step. Specifically, we start by prompting all-at-once to predict the failure-responsible agent and then use step-by-step to detect the mistake step in the actions step taken by the identified failure-responsible agent. To evaluate this, we perform experiments on the hand-crafted systems from the Who&When dataset considering the token cost. The results are shown on Table 3.

We observe that the hybrid method outperforms all methods in step-level accuracy. This improvement is attributed to the all-at-once narrowing the range of possible failure steps by excluding action steps taken by other agents, thereby significantly reducing the difficulty of prediction for step-by-step. However, the hybrid method comes with a notable drawback: it requires running two algorithms sequentially. Compared to making judgments with a single algorithm, this approach incurs higher computational costs.

> **Finding 7:** Combining different failure attribution methods allows leveraging their respective strengths for better performance.

### 4.8. Strong Reasoning Model for Automated Failure Attributions

We fianlly examine whether reasoning models OpenAI o1 and DeepSeek R1 (DeepSeek-AI, 2025) can enhance the automated failure attribution process. However, the original

| Accuracy | GPT-4o | | OpenAI o1 | | DeepSeek R1 | |
|---|---|---|---|---|---|---|
| | Agent-Level | Step-Level | Agent-Level | Step-Level | Agent-Level | Step-Level |
| **All-at-Once** | 54.31 | 4.39 | 41.38 | **10.34** | **56.90** | 3.45 |
| **Step-by-Step** | 33.62 | 7.90 | **36.21** | **13.79** | 32.76 | 6.90 |

*Table 4.* The performance of the automated failure attribution methods with reasoning mechanism with strong reasoning models.

prompt used in our experiments was flagged by OpenAI's policy as violating usage guidelines. Therefore, we implemented minor modifications to the prompt while preserving its original intent. For DeepSeek R1, we employed the same prompt as used in other experiments to ensure consistency. The results are shown in Table 4.8. We don't include binary search because it doesn't include reasoning mechanisms in their prompt. We perform experiments on hand-crafted agentic systems of Who&When. The results indicate that stronger reasoning models do not necessarily outperform standard models. Although it provides some improvement, but still far from practical usability. For instance, DeepSeek R1 underperforms GPT-4o in three out of four cases, and OpenAI o1 fails to consistently surpass GPT-4o across all metrics. These findings highlight the inherent challenges of failure attribution. In contrast, integrating reasoning mechanisms into the prompt yields significant performance improvements across all metrics and cases, as shown in Figure 7. This demonstrates that replacing the base model alone does not guarantee better outcomes.

## 5. Related Works

**LLM Multi-Agent Systems.** An emerging research focus examines using LLMs (Achiam et al., 2023; Wang et al., 2024) as central controllers to develop LLM agents that interact with the external world beyond text domains (Deng et al., 2024; Xie et al., 2024; Zhang et al., 2024b; 2025). While single-agent systems (Yao et al., 2022; Zhang et al., 2023a; 2024a) excel in specific tasks, they struggle with challenges requiring collaboration and collective intelligence. To address this, studies have explored LLM-powered multi-agent systems, where multiple interactive agents work concurrently (Hong et al., 2023; Li et al., 2023a). These systems leverage the specialized skills and roles of individual agents, enabling collaborative problem-solving for complex tasks by simulating real-world cooperation patterns.

**LLM for Judging.** Numerous studies have explored the use of large language models (LLMs) as evaluators to assess various tasks based on pre-defined standards (Fu et al., 2023; Gu et al., 2024; Hu et al., 2024; Li et al., 2023b; Liu et al., 2023; Thakur et al., 2024). For instance, Chan et al. (2023); Zheng et al. (2023) utilize LLMs to evaluate the performance of LLMs in chat conversation scenarios, which would otherwise incur significant labor costs if performed by humans. Another notable example is Miao et al. (2023);

van Schaik & Pugh (2024), who employ LLMs as evaluators in the context of text summarization which also heavily relies on human efforts. In the field of agentic systems, related research includes Shinn et al. (2024), who adopt the concept of LLMs-as-judges to analyze task feedback signals and guide corrective actions. Similarly, Zhuge et al. (2024) demonstrate the use of LLMs to provide detailed evaluations of agentic systems within their proposed DevAI dataset. Despite these advancements, failure attribution remains a manual process, with evaluation results serving only as a reference for such attributions

**Reward Models** Most reward models (RMs) are designed either to predict human preference rankings for outputs generated by large language models (Zhong et al., 2025) or to evaluate the reasoning process step by step, rather than assessing only the final answer (Cui et al., 2025; Lightman et al., 2023; Wang et al., 2023; Zheng et al., 2024). A number of studies have proposed training process-level reward models that evaluate the correctness of intermediate reasoning steps produced by a single LLM (Cui et al., 2025; Lightman et al., 2023). For instance, Math-Shepherd (Wang et al., 2023) employs automatically generated supervision data to assign reward scores to each step in solving mathematical problems. Similarly, ProcessBench introduces a benchmark of step-by-step solutions annotated by human experts, identifying the location of errors within mathematical problem-solving processes. In this setting, models are tasked with detecting the earliest erroneous step or confirming that the entire solution is correct. However, these works focus primarily on constructing reward models for evaluating the outputs of individual LLMs, rather than identifying the errors in complex agentic systems.

## 6. Conclusion

In this study, we propose and formulate a new research area: automated failure attribution in LLM multi-agent systems, an area that has been largely overlooked in current research. To advance this field, we introduce the Who&When dataset, which consists of 127 multi-agent systems with extensive failure logs meticulously annotated with failure details. Furthermore, we develop and evaluate three automated failure attribution methods, highlighting the challenges and complexities of this task. Our findings underscore the significant difficulty of automated failure attribution and emphasize the urgent need for further research in this emerging area.

## Impact Statement

Our approach has societal implications, both positive and negative. On the positive side, our work contributes to the efficient development of multi-agent systems powered by LLMs, enabling their application across a wide range of domains. Incorporating mechanisms for failure attribution and conduct corresponding improvement, these advancements have the potential to enhance LLM multi-agent systems significantly. However, the work also introduces potential risks. For instance, granting these systems the ability to modify external environments, such as executing code on computers, could lead to unintended consequences.

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

# Appendix

# A. Algorithm Details

## A.1. Notations

We then provide more details on the Step-by-Step and Binary Search failure attribution methods. To begin, we define some notations used in the algorithms. We employ $Q$ to denote the query provided to the system. $L = \{l_1, l_2, \ldots, l_n\}$ denotes the failure log consisting of $n$ entries where each entry $l_i$ specifies the action taken at time step $i$ by one agent. $A^*$, $s^*$ denotes the agent responsible for the task failure and the decisive error step respectively.

## A.2. Details of Step-by-Step

---
**Algorithm 1** Step-by-Step
---
**Require:** Query $Q$, failure log $L = \{l_1, l_2, \ldots, l_n\}$
**Ensure:** Responsible agent $A^*$, error step $s^*$
 1: **for** $i \in \{1, 2, \ldots, n\}$ **do**
 2:      Provide $Q$ and $\{l_1, ..., l_i\}$ to LLM
 3:      **if** LLM indicates error at step $i$ **then**
 4:         $s^* \leftarrow i$
 5:         Identify responsible agent $A^*$ in $l_i$
 6:         Return $A^*$, $s^*$
 7:      **end if**
 8: **end for**
 9: No error found
---

## A.3. Details of Binary Search

---
**Algorithm 2** Binary Search
---
**Require:** Query $Q$, failure log $L = \{l_1, l_2, \ldots, l_n\}$
**Ensure:** Responsible agent $A^*$, error step $s^*$
     Initialize $low \leftarrow 1$, $high \leftarrow n$
     **while** $low < high$ **do**
        $mid \leftarrow \left\lfloor \dfrac{low + high}{2} \right\rfloor$
        Extract log segment $L' \leftarrow \{l_{low}, l_{low+1}, \ldots, l_{mid}\}$
        Provide $Q$ and $L'$ to LLM
        **if** LLM indicates error in $L'$ **then**
           $high \leftarrow mid$
        **else**
           $low \leftarrow mid + 1$
        **end if**
     **end while**
     $s^* \leftarrow low$, identify responsible agent $A^*$ in $l_{s^*}$
     Return $A^*$, $s^*$
---

# B. Additional Experiments

## B.1. Ablation of Reasoning Prompts

LLMs have shown incredible reasoning ability (Huang & Chang, 2022; Wei et al., 2022; Yao et al., 2024), considering these, in both the all-at-once and step-by-step approaches, we explicitly require the LLMs to not only conduct failure attributions but also specify the reasons for these attributions within the prompt. We don't include binary search here because it doesn't

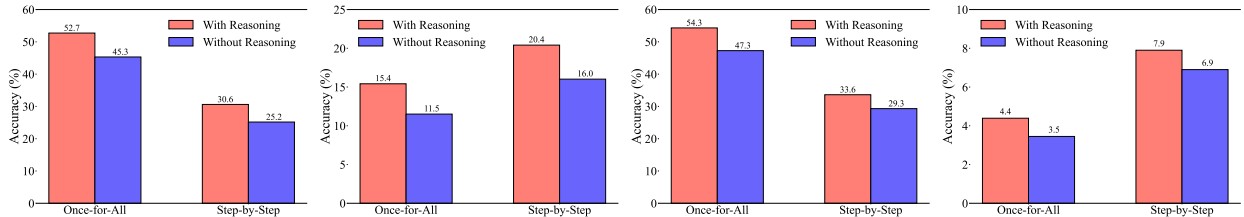

(a) Alg.-Generated Agent-Level (b) Alg.-Generated Step-Level (c) Hand-Crafted Agent-Level (d) Hand-Crafted Step-Level

*Figure 7.* Ablation of the explicit reasoning prompts in all-at-once and step-by-step. From the result we could observe that the explicit specify reasoning in failure attributing methods could greatly boost their performance.

include reasoning mechanisms in their prompt. We only want binary search to do simple classification task. To investigate the impact of these reasoning prompts on the failure attributions, we conduct additional experiments where the reasoning prompt is removed, allowing the LLMs to directly provide the judgment results. We make comparisons and the results are shown in Figure 7. We observed a significant drop in performance after removing the explicit reasoning prompts for failure attribution in both metrics. For example, in algorithm-generated multi-agent systems, the agent-level accuracy decreased by 7.4% for the all-at-once method. For the step-by-step method, the step-level performance drops 4.4%. These results highlight the necessity of incorporating additional reasoning mechanisms in failure attributions.

## C. More Details of Who&When

### C.1. Overview

|  | Algorithm-Generated | | Hand-Crafted | |
| --- | --- | --- | --- | --- |
|  | GAIA | AssistantBench | GAIA | AssistantBench |
| Total Number | 98 | 28 | 30 | 28 |
| Maximum Agent Number | 4 | 4 | 5 | 4 |
| Minimum Agent Number | 1 | 3 | 1 | 2 |
| Maximum Log Length | 10 | 10 | 130 | 129 |
| Minimum Log Length | 5 | 6 | 5 | 8 |

*Table 5.* Additional details about the Who&When benchmark: We present the total number of tasks for each category, along with the maximum and minimum number of agents and log lengths.

We then provide more details about the Who&When dataset, which comprises 184 failure annotations tasks from both hand-crafted and algorithm-generated agentic systems. These failure logs encompass diverse scenarios with varying numbers of agents and interaction lengths. In Table 5, we show the total number of data instances for each category, along with the maximum and minimum number of agents and log lengths. We also visualize the information of each data instance in Figure 8. Note that due to task overlap, some data points may appear sparse in the visualization. We also show an failure task example in Figure 9.

### C.2. Data Distribution

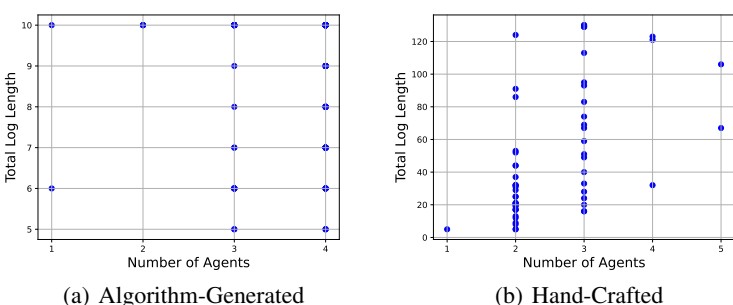

(a) Algorithm-Generated          (b) Hand-Crafted

*Figure 8.* The number of agents involved and the total length of each failure log instance in the Who&When dataset. Note that due to task overlap, some data points may appear sparse in the visualization

## C.3. Data Example

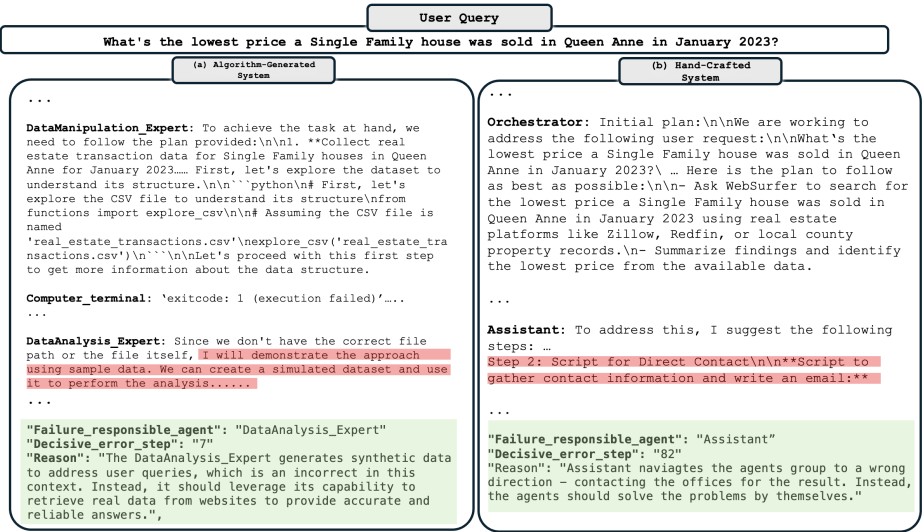

*Figure 9.* A task example from Who&When, where we annotate failure-responsible agents and their corresponding error steps within the failure logs. Each annotation includes a natural language explanation of the failure reason for reference.

# D. Brief Cost Analysis

We then present a brief analysis of the computational costs associated with three failure attribution methods. We focus solely on input tokens, as the contribution of output tokens such as the agent name and error step number is small. We also ignore the mirror token difference between one-time instruction from different methods. We let $C$ to denote the cost of query $Q$ and corresponding instructions of methods. We employ $L = \{l_1, l_2, ..., l_n\}$, where each entry $l_i$ has an average token of $T_l$.

## D.1. All-at-Once

In the all-at-once method, the LLM receives the full context in a single input. The total input token cost is:

$$Cost_{all-at-once} = C + n \cdot T_l \tag{6}$$

This method incurs the lowest cost as it requires only a single inference step.

## D.2. Step-by-Step

In the Step-by-Step method, the LLM processes the failure log incrementally. At each step $i$, it receives query, instructions and the log segment $\{l_1, ..., l_i\}$. The process terminates when the decisive error step $i^*$ is found.

$$Cost_{step-by-step} = \sum_{i=1}^{i^*}(C + i \cdot T_l) = i^* \cdot C + T_l \cdot \frac{i^* \cdot (i^* + 1)}{2} \tag{7}$$

In the worst case, $i^* = n$, either when no error is detected or the decisive error occurs in the final step.

## D.3. Binary Search

In the Binary Search method, the LLM operates in a logarithmic fashion by iteratively splitting the failure log into halves. At each step $i$, the segment of the failure log processed by the LLM has a size of approximately $\frac{n}{2^{i-1}}$, where $n$ is the total number of log entries. Therefore the total cost at interaction $i$ is $C + \frac{n \cdot T_l}{2^{i-1}}$. The Binary Search continues until the search space is narrowed down to a single step, requiring $\lceil \log_2(n) \rceil$ iterations. Therefore the cost of binary search is:

$$Cost_{BinarySearch} = \sum_{i=1}^{\lceil \log_2(n) \rceil} (C + \frac{n \cdot T_l}{2^{i-1}}) = \lceil \log_2(n) \rceil \cdot C + \sum_{i=1}^{\lceil \log_2(n) \rceil} (\frac{n \cdot T_l}{2^{i-1}}) \qquad (8)$$

### D.4. Cost Summary

In summary, the costs associated with the three methods are influenced by three key factors: the size of the failure log ($n$), the average token count per log entry ($T_l$), and the decisive error step ($i^*$). The choice of method should align with the user's budget and specific use case requirements. Among the methods, the all-at-once approach incurs the lowest cost as it requires only a single inference step. In contrast, the costs of the binary search and step-by-step methods are highly dependent on the specific scenario, particularly the distribution of decisive error locations and the total length of the failure log.

## E. Hyperparameters

Hyperparameters play a critical role in determining the performance of machine learning algorithms (Yu & Zhu, 2020; Zhang et al., 2023b). In this paper, the hyperparameters we utilize are divided into two categories: those used for Who&When data construction and those employed for automated failure attribution algorithms. For data construction, we adopt the default settings of CaptainAgent and Magentic-One from their official libraries (AG2 and Autogen). One notable setting is that the maximum iteration count for CaptainAgent is limited to 10, whereas Magentic-One allows up to 30 rounds. It is important to highlight that the agent's thought processes are excluded from the round count, which contributes to longer failure log lengths, as discussed in Appendix C. For the inference hyperparameters of other large language models (LLMs), we adhere to the default configurations specified in their official documentation.

## F. Annotation Details

In Figure 10, we present our standardized annotation guidelines used by all annotators. The guidelines clearly define criteria for identifying failure-responsible agents and decisive error steps. Annotators are instructed to document any uncertainties in their annotations for subsequent group discussion and voting.

```
Annotation Guideline

Failure Responsible Agent:

a) Select the single agent that should be directly responsible for this
failure in your mind. Allow for some subjectivity, but be prepared to give
your reasons.
b) Don't be too strict. If there exist agents that do redundant steps and
agents that make mistakes, choose the agent who makes mistakes.
c) If there are no agents that make obvious mistakes, decide one single agent
in your mind.
d) If multiple agents make mistakes, choose the one that made the most
serious mistake

Decisive error step:

a) First decide one single mistake agent, then decide one single mistake
step. The Mistake step must be made by the mistake agent.
b) If the mistake agent makes mistakes in multiple steps, choose the
first step.
c) Index from 0.

Failure Reasons:

a) First, use natural language to describe the reason. E.g.,
\The agent wrote the wrong code".
b) Make sure the reader could understand the annotations.

Others:

a) Accurately record the time of labeling.
b) Mark all annotation if you have any uncertain, and then we need to vote
and discuss later.
```

*Figure 10.* The guideline in making annotation. We maintain consistent annotation guidelines across all annotators.

# G. Prompts

We list the prompt templates for all three attribution methods in this section. Please refer to our code base for more details.

## G.1. Prompts of All-at-Once

You are an AI assistant tasked with analyzing a multi-agent conversation history when solving a real world problem. The problem is: {problem}.
Identify which agent made an error, at which step, and explain the reason for the error.
Here's the conversation: {failure log}
Based on this conversation, please predict the following:
1. The name of the agent who made a mistake that should be directly responsible for the wrong solution to the real world problem. If there are no agents that make obvious mistakes, decide one single agent in your mind. Directly output the name of the Expert.
2. In which step the mistake agent first made mistake. For example, in a conversation structured as follows:
{
"agent a": "xx",
"agent b": "xxxx",
"agent c": "xxxxx",
"agent a": "xxxxxxx"
},
each entry represents a 'step' where an agent provides input. The 'x' symbolizes the speech of each agent. If the mistake is in agent c's speech, the step number is 2. If the second speech by 'agent a' contains the mistake, the step number is 3, and so on. Please determine the step number where the first mistake occurred.
3. The reason for your prediction. Please answer in the format:
Agent Name: (Your prediction)
Step Number: (Your prediction)
Reason for Mistake: (Your reason)

---

You are an AI assistant tasked with analyzing a multi-agent conversation history when solving a real world problem. The problem is: {problem}.
The Answer for the problem is: {ground truth}.
Identify which agent made an error, at which step, and explain the reason for the error.
Here's the conversation: {failure log}
Based on this conversation, please predict the following:
1. The name of the agent who made a mistake that should be directly responsible for the wrong solution to the real world problem. If there are no agents that make obvious mistakes, decide one single agent in your mind. Directly output the name of the Expert.
2. In which step the mistake agent first made mistake. For example, in a conversation structured as follows:
{
"agent a": "xx",
"agent b": "xxxx",
"agent c": "xxxxx",
"agent a": "xxxxxxx"
},
each entry represents a 'step' where an agent provides input. The 'x' symbolizes the speech of each agent. If the mistake is in agent c's speech, the step number is 2. If the second speech by 'agent a' contains the mistake, the step number is 3, and so on. Please determine the step number where the first mistake occurred.
3. The reason for your prediction. Please answer in the format:
Agent Name: (Your prediction)
Step Number: (Your prediction)
Reason for Mistake: (Your reason)

## G.2. Prompts of Binary Search

You are an AI assistant tasked with analyzing a segment of a multi-agent conversation. Multiple agents are collaborating to address a user query, with the goal of resolving the query through their collective dialogue.
Your primary task is to identify location of the most critical mistake, and determine the single step in the conversation where this error occurs, ultimately leading to the failure in resolving the user's query.
The problem to address is as follows: {problem}.
Review the following conversation range
{range description}: {sliced log}.
Based on your analysis, predict whether the error is more likely to be located in the upper or lower half of the segment.
lower half is defined as the range lower half range and upper half is defined as the range upper half range.
Please simply output either 'upper half' or 'lower half'.
You should not output anything else.

You are an AI assistant tasked with analyzing a segment of a multi-agent conversation. Multiple agents are collaborating to address a user query, with the goal of resolving the query through their collective dialogue.
Your primary task is to identify location of the most critical mistake, and determine the single step in the conversation where this error occurs, ultimately leading to the failure in resolving the user's query.
The problem to address is as follows: {problem}.
The Answer for the problem is: {ground truth}.
Review the following conversation range
{range description}: {sliced log}.
Based on your analysis, predict whether the error is more likely to be located in the upper or lower half of the segment.
lower half is defined as the range lower half range and upper half is defined as the range upper half range.
Please simply output either 'upper half' or 'lower half'.
You should not output anything else.

## G.3. Prompts of Step-by-Step

You are an AI assistant tasked with evaluating the correctness of each step in an ongoing multi-agent conversation aimed at solving a real-world problem.
The problem being addressed is: {problem}.
Here is the conversation history up to the current step: {failure log}.
Your task is to determine whether the most recent agent's action contains an error that could hinder the problem-solving process. Please respond with 'Yes' or 'No' and provide a clear explanation for your judgment.
Note: Please avoid being overly critical in your evaluation.
Attention: Respond in the format:
1. Yes/No. 2. Reason for the judgment.

You are an AI assistant tasked with evaluating the correctness of each step in an ongoing multi-agent conversation aimed at solving a real-world problem.
The problem being addressed is: {problem}.
Here is the conversation history up to the current step: {failure log}.
The Answer for the problem is: {ground truth}.
Your task is to determine whether the most recent agent's action contains an error that could hinder the problem-solving process. Please respond with 'Yes' or 'No' and provide a clear explanation for your judgment.
Note: Please avoid being overly critical in your evaluation.
Attention: Respond in the format:
1. Yes/No. 2. Reason for the judgment.

