# OpenReview forum: "Which Agent Causes Task Failures and When? On Automated Failure Attribution of LLM Multi-Agent Systems"
_ICML.cc/2025/Conference — ICML 2025 spotlightposter_

### Official Review · Reviewer_micc · 2025-03-09

**Overall Recommendation:** 4

**Summary:**

The paper explores automated failure attribution in LLM multi-agent systems. It introduces and formulates a new research of identifying the agent and specific step responsible for task failures within agentic systems. The research introduces the Who&When dataset, which contains failure logs from 127 LLM multi-agent systems, annotated to link failures to particular agents and error steps.
The paper evaluates three automated failure attribution methods, demonstrating their strengths and limitations. The best method achieved 53.5% accuracy in identifying failure-responsible agents but only 14.2% in pinpointing failure steps, underscoring the complexity of the task.
The authors argue that evaluation and failure attribution should be integrated and that more effort is needed to bridge the gap between evaluation results and failure attribution. They propose leveraging LLMs for automated failure attribution to reduce the need for manual analysis and enable human resources to focus on improving system functionality.
Overall, the findings highlight the challenges of using LLMs for failure analysis in multi-agent systems and the need for further research in this area.

# update after rebuttal I maintain my score of Accept and think authors discussed an important and interesting topic with proper experimental approach.

**Claims And Evidence:**

In general, the paper is well structured and written, and most of the claims are logical and sufficiently substantiated.
Main claims are:
1) Automated failure attribution for multi-agent LLM systems is underexplored and novel, but it is important for debugging purposes. Especially with increasing complexity of these systems.

Their claim is supported by literature overview demonstrating reliance on human labor that is resource-intensive. The authors further support it with the empirical results of human hours involved in data labelling.

2) A new dataset is constructed and annotated containing logs from 127 multi-agent systems  with 184 failure annotation tasks claiming to advance the research in this area.

The dataset indeed is thoroughly analysed and used to provide three methodologies applied to it.

3) Current three automated methods achieve modest accuracy (53.5% accuracy at identifying agents responsible for failure and 14.2% at pinpointing exact step), highlighting significant complexity in automation of this task.

The experimental evidence is clear, it thoroughly outlines findings that can be logically followed.

Somehow problematic claims:
The paper explores a relatively new research area, supported by a newly introduced dataset, which contains annotations from 3 annotators and from a single hand-crafted agent system and multiple automatically generated agentic systems. Several findings exclude logs from automated ones (with well justifiable notes but still). This limits the diversity and generalisation raising questions if these results will hold for a broader variety of agentic systems. Saying that, I still think it is important to raise issues and problems but expressing those limitations in the paper will help to caveat it for the audience.

The second issue worth raising is the obvious subjectivity in annotation. While authors raise that the consensus was built after, it highlights that the 'ground truth' might be noisy and further highlights that the accuracy results can be unreliable.

**Essential References Not Discussed:**

In my opinion, all the required references are there contributing to a structured and logical flow.

**Experimental Designs Or Analyses:**

The experimental designs and analyses are mostly  sound and valid for the problem at hand.

There are concerns for representativeness of accuracy due to limited data and annotators disagreement (only three people).

A few suggestions and comments:
	1) The paper does not clearly explain how the random baseline was constructed. Detailed information on this would enhance understanding and evaluation.
	2)  Figure 8 appears to have non-integer values for the number of agents ( we can't expect to have 1.5 or 4.5 agents). Adjusting the x-axis to display only integer values would be more appropriate.
	3) I was puzzled by Annotation guideline (Figure 10) point c). Does it force to choose agent even when none can be found by an annotator?  This can potentially lead to higher levels of uncertainty and some annotators in the debate biasing the rest.
	4) In Section 2 the formulation of the problem defines a turn-based protocol. While it can be seen as a problem scope for this paper, an introduction in mathematical formulation sounds like this is the only way LLM agents can act, which is not true. Clearly specifying that this is the scope for this paper will make it more clear.

**Methods And Evaluation Criteria:**

Overall, the approaches and metrics used in the study are appropriate for addressing the complexities and challenges of failure attribution in LLM multi-agent systems. While the results are quite modest, I appreciate the complexity and documentation of those weak results.

One issue in the appendix discusses computational costs for various failure attribution methods as mathematical function of the size of the failure log, the token count and error step. However, providing empirical results that demonstrate these costs in practice would substantiate these theoretical calculations. Just as the hybrid method's token expense is discussed in a practical context in the paper, showing how these costs manifest in experiments would be helpful.

**Other Comments Or Suggestions:**

Here is a list of various typos and unclear sentences:
	1) Page 1 line 039, second column: manual efforts involve(s)
	2) Couldn't find a reference to Figure 1 in the text
	3) Page 4 lines 180-181: "We exclude the entire GAIA dataset" - but prior to that you say that you use randomly sampled instances of GAIA. It is confusing. Do you mean you exclude the rest of GAIA dataset?
	4) Page 4, lines183-184, second column: "for both normal people and domain experts" :) It assumes that domain experts are abnormal. Suggest to change it to non-experts.
	5) Page 5, line 221, first column: "we are thinking of performing" - assuming the rest is in past tense, it would be better to keep it as past tense. After all, you have already reported it.
	6) Page 12 Figure 7 description "the explicit specify reasoning"
	7) Page 12 line 630, "We only want(s)" - same in line 653
	8) Page 12 line 634 "attribution in both ~two~ metrics"
	9) Page 12 lines 655 "Although may provide some improvement<…> - reads like a comment, needs to be a full sentence.
	10) Page 13 line 665 'specify" -> specifies
	11) Figure 8 x-axis does not make sense with non-integer values
Figure 10 Others b) needs rephrasing and checking.

**Other Strengths And Weaknesses:**

The paper presents notable strengths in focusing on a new and increasingly critical area of research—automated failure attribution in LLM multi-agent systems. The growing complexity of agentic workflows means that effective debugging is essential. The authors have taken a significant first step by identifying this problem and introducing a new dataset and a few methods designed to address failure attribution. The currently weak performances of the proposed methods only further highlight the complexity of the task. The empirical evaluations of the All-at-Once, Step-by-Step, and Binary Search methods pave the way for future advancements and provide a crucial foundation for ongoing research.

The paper also has its weaknesses. The constructed dataset is somewhat limited, which may restrict the generalizability of the results. The accuracy of the proposed methods is primarily based on this limited data and involves only a few annotators, leading to potential issues with the reliability and validity of the ground truth. This limited scope could result in higher levels of uncertainty and potential biases in the findings. To ensure more robust and widely applicable results, future work should consider expanding the dataset and involving a more extensive and diverse pool of annotators, agentic designs and logs to establish clearer and more certain ground truth data and robust findings.

**Questions For Authors:**

No other questions

**Relation To Broader Scientific Literature:**

The paper contributes to the broader scientific literature by addressing the challenge of automated failure attribution in large language model (LLM) multi-agent systems. Here are the key contributions and their relation to prior findings and ideas:

1. Automated Failure Attribution aspect:
 Prior research has largely focused on using LLMs for various evaluation tasks, leveraging LLMs to reduce human labor. The paper introduces the concept of automated failure attribution within LLM multi-agent systems, proposing methods to identify the agent and steps responsible for failures. This extends the application of LLMs from evaluation to a more diagnostic  role.

2. Who&When Dataset construction:
Existing datasets often focus on evaluating model outputs in isolation without detailed annotations linking failures to specific agents and steps (like DevAI and SWE-Bench). The Who&When dataset fills this gap by providing failure logs from 127 multi-agent systems, with annotations that link failures to specific agents and errors. While this dataset supports more detailed and systematic failure analysis than previously possible, it still lacks of comprehensiveness. It can be better positioned as a first step towards automated failure attribution rather than finalised dataset that can be used to solve the problem.

3. Evaluation of Failure Attribution Methods:
 Manual failure attribution has been the norm (Gu et al., 2024; Tan et al., 2024; Zheng et al., 2023), with a significant labor cost and potential for human error. This paper evaluates three automated methods—All-at-Once, Step-by-Step, and Binary Search—demonstrating their respective advantages and limitations in identifying failure-responsible agents and steps. This empirical evaluation provides a first step towards understanding of the effectiveness of these methods, offering a benchmark for future improvements.

**Theoretical Claims:**

The paper primarily focuses on empirical evaluations and practical methodologies rather than providing detailed theoretical proofs. Its main contributions involve introducing the Who&When dataset and evaluating three methods for automated failure attribution, highlighting their strengths and weaknesses through empirical results.

Given this focus, the paper does not present theoretical proofs that require verification for correctness. Instead, it relies on experimental results to support its claims. The evaluation metrics and empirical analyses serve as the basis for the paper's conclusions.

Therefore, there are no theoretical proofs within the paper that need to be checked for correctness.

Saying that, there is a problem foundation that introduces mathematical notations for automated failure attribution that I thoroughly checked.

One comment: the paper indeed introduces mathematical notations, however, it's practical utility in the paper is unclear, given that the authors almost do not utilise it for any theoretical derivations. The notation appears primarily illustrative rather than operational.

---

> ### Author Rebuttal · Authors · 2025-03-30
>
> We sincerely appreciate your valuable time and insightful feedback! We address each of your questions in our responses below.
>
> **[Re Comment 1 on Claims & Comment 4 on Experiments: Need to specify the scope. ]**
>
> We will incorporate all your suggestions and make the following clarifications and revisions in future versions, Specifically:
> We will explicitly state at the beginning of Section 2 that the initial failure attribution task targets turn-based LLM multi-agent systems, aligning with prior work and widely adopted agentic frameworks [1][2]. We will also emphasize that the protocol is rigorously defined, as detailed in the Background of Section 2.
> To better inform readers about the scope and diversity of Who&When, we will expand the statistical overview in Section 3. This will include the total number of agents, a summarization of entire action steps, and detailed tools information—facilitating better adaptation to specific use cases.
>
> [1] Wu, Qingyun, et al. "Autogen: Enabling next-gen llm applications via multi-agent conversation."
> [2] Li, Guohao, et al. "Camel: Communicative agents for" mind" exploration of large language model society."
>
> **[Re Comment 2 on Claims:  The subjectivity in annotation.]**
>
> The disagreement number reported in Figure 2 reflects only the initial stage of annotation. Annotators are required to reach a consensus by strictly adhering to the problem formulation in Section 2 rather than relying on subjective judgment. During debating, expert annotators must justify their positions based solely on this formal definition, ensuring precision and unambiguousness. We will clarify this in the revised Section 3.2.
>
> **[Re Comment 1 on Methods: Need to provide empirical results that demonstrate the cost calculations.]**
>
> Thank you for highlighting this detail! We have provided empirical cost calculations for all methods in Table 3. The results indicate that the Hybrid Method incurs the highest cost, followed by Step-by-Step, Binary Search, and All-at-Once. We would be happy to discuss them if you have further questions!
>
> **[Re Comment 1 on Theory: The practical utility of mathematical notations.]**
>
> The mathematical notation introduced in Section 2 aims to rigorously define failure-responsible agents and decisive error steps. These definitions (1) enable readers to clearly understand the research problem without ambiguity, and (2) establish strict guidelines for subsequent annotations in constructing the Who&When dataset.
>
> **[Re Comment 1 on Experiments:  Details of Random search.]**
>
> The random baseline is constructed using a uniform selection strategy over all available options. Step-level accuracy is defined as the inverse of the average number of steps per problem, and agent-level accuracy as the inverse of the number of agents per problem. Final baseline performance is obtained by averaging these values across all data. We will include this explanation in the revised manuscript.
>
> **[Re Comment 2 on Experiments:  Adjusting the x-axis in Fig. 8.]**
>
> Thank you for the suggestion. We will adjust the x-axis in the revised manuscript by removing non-integer values for clarity.
>
> **[Re Comment 3 on Experiments: Does it force annotators to choose an agent even when none can be found?]**
>
> Yes. We force annotators to choose one agent with their best rather than passively accepting others’ opinions; we also explicitly require annotators to highlight any uncertain annotations as shown in guideline (b). This helps maintain accuracy and eliminate biases. We will clarify this intention further in our revised manuscript.
>
> **[Re Weakness 1 & Suggestion: The constructed dataset is somewhat limited & future work should consider expanding the dataset.]**
>
> Thank you for the constructive suggestions! A key direction for future work is expanding the dataset to further benefit the research community. We also kindly emphasize that Who&When already covers a substantial scope: it includes 127 LLM-based multi-agent systems, 201 agents using 48 tools, and 4,092 action steps. The annotation process required 84.3 hours of expert effort across three annotators.
>
> **[Re Other Comment:]**
>
> We sincerely appreciate your detailed feedback and will incorporate all your suggestions. Specifically:
>
> **(1), (5), (7), (8):** We acknowledge the typographical errors and will correct verb tenses, third-person singular usage, and the redundancy in "both two."
>
> **(2):** We will add explicit references to Figure 1 at lines 48 and 70 when introducing "manual failure attributions" and "automated failure attribution."
>
> **(3), (4), (6)** We will revise all these unclear phrasings in accordance with your recommendations!
>
> **(9):** We will rephrase for clarity: "Although strong reasoning models yield improvement on the Who&When dataset, their performance remains insufficient for practical use in failure attribution."
>
> **(10):** We will remove non-integer x-axis values and revise the instruction to: "Record all uncertain annotations."

---

> > ### Comment · Reviewer_micc · 2025-04-06
> >
> > Thanks for the response. I really like this paper and maintain the current rating as it is at 'Accept' level already but I believe all of the improvements will make this paper more solid.

---

> > > ### Author Response · Authors · 2025-04-07
> > >
> > > We sincerely appreciate your valuable time and many constructive comments. Thank you for highlighting the strengths of our work! We will incorporate all your suggestions into our next version accordingly!

---

### Official Review · Reviewer_Fdny · 2025-03-10

**Overall Recommendation:** 4

**Summary:**

This paper introduces automated failure attribution for LLM-powered multi-agent systems, addressing the problem of identifying which agent causes a task failure and at which step the decisive error occurs. The authors formally define this research area, propose Who&When, a dataset with annotated failure logs from 127 LLM-powered multi-agent systems, and evaluate three automated attribution methods. Their experiments show significant performance variability related to context length and model choice, highlighting that current LLM-based methods are not yet practically usable and underscoring the need for further research.

**Claims And Evidence:**

(1) Problem Establishment: The authors rigorously formalize and motivate the automated failure attribution problem in LLM-powered multi-agent systems (Section 2). (2) Key Findings: The extensive experiments in Section 4 yield seven important insights, such as the relationship between context length and failure attribution performance. To me, these insights meaningfully guide future advancements in the field.

**Essential References Not Discussed:**

N/A

**Experimental Designs Or Analyses:**

I have discussed the experimental designs in the previous parts. I like the finds listed in experimental section like the correlation between context length and failure attribution performance.

**Methods And Evaluation Criteria:**

The authors mainly investigate three automated failure attribution methods they proposed in this work on the Who\&When benchmark. Additionally, they also proposed two main metrics they established in this area, i.e, agent-level accuracy and step-level accuracy. The benchmark, the evaluation metrics and the methods they proposed makes senses to me and well-motivated.

**Other Comments Or Suggestions:**

1. Experiments in Section B.2 should be incorporated in main experiments but not Appendix. Reasoning models are also LLMs and it should be compared with other models in Figure 3.
2. The experiments in Section B.2 are conducted on subset of  Who&When or the entire dataset? I am not able to find any information about this on the paper.

**Other Strengths And Weaknesses:**

**Strengths**

(1) The paper is overall well-structured and well written. The explanations are easy to follow and the logic flows are reasonable. The quality of the paper is very good.

(2) The problem itself is very interesting and important. The problem formulation is mostly clear (see comments before) and the authors provide intuition about how it this problem is important. I am convinced by these arguments.

(3) The authors propose the first benchmark named Who&When for failure attributions with established metrics. These metrics are well-motivated and makes senses to me. I have also checked the anonymous repository associated with the paper. The dataset quality is great and the annotations are also in a high-standard.

(4) The authors perform extensive experiments on the proposed benchmarks and the conclusions are meaningful and could potentially guide the research in this new area.

**Weakness**

(1) Why not making a comparison/ do some analysis with Agent-as-Judge in your experimental section?  If I understand correctly, the agent-as-judge could also be applied to this area with some mirror modifications.

(2)  Surprisingly, I see DeepSeek R1 performs worse than GPT-4o in step-level accuracy, while OpenAI O1 model is worse than GPT-4o in Agent-Level accuracy. Does the authors have some explanations for that? Some discussion needed to be included in the paper. If these conclusions hold, do you have some plan to incorporate model selection in the failure attribution procedures?

(3) Different methods seem to have different advantages. For example,  all-at-once is good at picking up mistake agent and step-by-step is good at picking up mistake step. Why not combining them?

**Questions For Authors:**

(1) Why the reasoning models perform worse than GPT-4o? Does the author have some explanation for this?

(2) The experiments in Section B.2 are conducted on subset of the Who&When datset or the entire dataset? I could not find any information about that on the paper.
(3) Why not performing experiments on the entire Who&When on all models in the experimental section? The experiments setting should be consistent.

**Relation To Broader Scientific Literature:**

N/A

**Theoretical Claims:**

The authors’ theoretical analysis are mostly around problem formulation and the cost analysis in Appendix. The problem formulation makes senses and rigorous. I also checked the Appendix and the conclusion are also correct.

---

> ### Author Rebuttal · Authors · 2025-03-29
>
> We sincerely thank the reviewer for the insightful comments! Please find our response to your comments below.
>
> **[Re Weakness 1: Why not make a comparison/do some analysis with Agent-as-Judge in your experimental section?]**
>
> We appreciate this suggestion; however, the research objective of our failure attribution task significantly differs from that of Agent-as-Judge, making a direct comparison unsuitable. Specifically:
>
> - **Distinct Research Objectives:** Agent-as-Judge employs LLM agents primarily to evaluate the completeness and quality of coding outputs generated by LLMs. Its main focus is on assessing delivered coding projects rather than identifying errors within agentic systems themselves.
>
> - **Different Target Objects:** In Agent-as-Judge, the evaluation targets the final outputs (coding projects) produced by LLMs. Conversely, our failure attribution research specifically examines agentic systems, explicitly aiming to pinpoint the responsible agents leading to failures and identify the exact decisive-error step.
>
> Given these substantial differences, we have not incorporated Agent-as-Judge into our current experimental setup. We will elaborate further  on these differences  in the related work section.
>
> **[Re Weakness 2 & Question 1: Need more explanations for Table 4 results.]**
>
> Thank you for the suggestion. We would like to clarify that the most strong reasoning model (OpenAI O1) still achieves the best performance on three out of four metrics, as highlighted in bold in the original table. However, we do observe instances where the reasoning model performs worse than GPT-4o. This could be attributed to several factors: reasoning models excel at tasks requiring complex logical inference, such as challenging mathematical problems, but they are not necessarily optimized for parsing intricate logs or identifying errors in complex agent-based systems, as these tasks fall outside their training objectives. Developing effective failure attribution methods is an important avenue that merits further research.
>
> **[Re Weakness 3: Why not combine different failure attribution methods? ]**
>
> We would also like to clarify that we did conduct experiments combining different failure attribution methods; these results are presented in Table 3. Our findings indicate that the Hybrid Method indeed achieves the best performance across both metrics.Thanks for the feedback!
>
> **[Re Weakness 4: Experiments in Section B.2 should be incorporated in main experiments but not Appendix.]**
>
> Thanks for the suggestions. We will consider moving it to the main paper after all additional experiments are done for these models.
>
> **[Re Suggestion 2 & Question 2: The experiment's setup.]**
>
> We conducted these experiments using the same experimental setup as described in the ablation studies in Section 4.4. Due to cost considerations, we did not use the complete dataset for these experiments. Thank you for your suggestion; we will clarify this point further in future revisions.

---

> > ### Comment · Reviewer_Fdny · 2025-04-03
> >
> > Thanks for the response. I like this work for the research area it explores. I will maintain my original rating.

---

> > > ### Author Response · Authors · 2025-04-03
> > >
> > > Thank you very much for your acknowledgment and valuable suggestions. We will incorporate these revisions into our next manuscript accordingly!

---

### Official Review · Reviewer_BgjX · 2025-03-12

**Overall Recommendation:** 3

**Summary:**

The paper introduces a new research problem of automated fault attribution in multi-agent systems. The task includes identifying both the agent and the corresponding step that lead to task failure. To study this task, a new benchmark dataset called Who&When is created by manually labeling 127 failure logs. Three prompting-based approaches are proposed and evaluated on the benchmark. Experimental results indicate that even SOTA LLMs struggle on this task.

**Claims And Evidence:**

The main claim made in the paper is that a new task for automated failure attribution is proposed, which is important and challenging. Their experimental results with SOTA LLMs highlight the challenging nature of this task. However, there are some issues with the task.

There is no discussion of existing work on verifiers, particularly process reward models (PRM)[1, 2], that also aim to identify errors at the step-level. How is the proposed task different from this line of work? I understand that the proposed task is to identify the root-cause step that caused failure, rather than “all” erroneous steps, as done in existing work. But, can existing methods and datasets [3] least be leveraged for this task? For example, a simple baseline could be to choose the earliest erroneous step identified by PRM.

            The other sub-task is to identify the failure-responsible agent. However, once the decisive error

step is identified, identifying the corresponding agent becomes trivial.

       b.  The current task formulation seems highly subjective as shown in Figure 2, with up to 50%

disagreement between human annotators. The authors mention that consensus was reached via discussions. But, is the task well-defined or does is a different task design required, such as choosing a set of problematic steps instead of a single one? Without this understanding, the practical utility of the current task and benchmark data remains questionable.

[1] Lightman, H., Kosaraju, V., Burda, Y., Edwards, H., Baker, B., Lee, T., ... & Cobbe, K. (2023, May). Let's verify step by step. In The Twelfth International Conference on Learning Representations.

[2] Wang, P., Li, L., Shao, Z., Xu, R. X., Dai, D., Li, Y., ... & Sui, Z. (2023). Math-shepherd: Verify and reinforce llms step-by-step without human annotations. arXiv preprint arXiv:2312.08935.

[3] Zheng, C., Zhang, Z., Zhang, B., Lin, R., Lu, K., Yu, B., ... & Lin, J. (2024). Processbench: Identifying process errors in mathematical reasoning. arXiv preprint arXiv:2412.06559.

**Essential References Not Discussed:**

Existing work on identifying step-level errors is missing, even though main task proposed in the paper is to identify the decisive failure step.



[1] Lightman, H., Kosaraju, V., Burda, Y., Edwards, H., Baker, B., Lee, T., ... & Cobbe, K. (2023, May). Let's verify step by step. In The Twelfth International Conference on Learning Representations.

[2] Wang, P., Li, L., Shao, Z., Xu, R. X., Dai, D., Li, Y., ... & Sui, Z. (2023). Math-shepherd: Verify and reinforce llms step-by-step without human annotations. arXiv preprint arXiv:2312.08935.

[3] Zheng, C., Zhang, Z., Zhang, B., Lin, R., Lu, K., Yu, B., ... & Lin, J. (2024). Processbench: Identifying process errors in mathematical reasoning. arXiv preprint arXiv:2412.06559.

**Experimental Designs Or Analyses:**

Figure 5 error bars are too large to draw any meaningful conclusion.

Were the results averaged over multiple LLM runs? What is the variance across runs?

**Methods And Evaluation Criteria:**

Given the high disagreement between annotators, some more discussion around reliability of the ground truth data would be useful. Why exactly is the task challenging? Are the disagreements between annotators usually within a range of steps?

The dataset size is quite small.

Proposed “step-by-step” method seems to assume that the agent cannot correct its actions.  The high performance achieved by this method could suggest that the annotated logs do not contain self-correction steps, even though it is a popular strategy [4]. A more diverse benchmark dataset containing such LLM strategies would help draw more general conclusions.

[4] Pan, L., Saxon, M., Xu, W., Nathani, D., Wang, X., & Wang, W. Y. (2023). Automatically correcting large language models: Surveying the landscape of diverse self-correction strategies. arXiv preprint arXiv:2308.03188.

**Other Comments Or Suggestions:**

Lines 036-040 and 047-059 are weirdly phrased

**Other Strengths And Weaknesses:**

Strengths

The problem is well motivated

Extensive analyses were conducted. Analysis and findings on consistency of the performance of proposed methods across LLMs is particularly useful.

For weaknesses, please refer to all issues above mainly around task formulation, dataset reliability and generality.

**Questions For Authors:**

How often is the decisive error step same as the earliest error made by any agent?

What length values do the 5 levels in Figure 4 correspond to?

**Relation To Broader Scientific Literature:**

The paper builds upon existing work on identifying step-level errors in LLM execution logs, and extends it to pinpoint the most severe error that led to failure.

Proposed LLM-as-judge approaches, such as binary search, could potentially be useful for other long-text evaluation tasks.

**Theoretical Claims:**

Cost analysis in Appendix E is useful and correct.

---

> ### Author Rebuttal · Authors · 2025-03-30
>
> We sincerely appreciate your valuable time and insightful feedback! Due to text limit, additional experiments are shown in anonymous link: **https://shorturl.at/JSeJd**.
>
> **[1. Discussion needed for verifiers & Can [3] be leveraged?]**
>
> Thank you for the suggestions! We acknowledge the necessity of discussing verifiers. We will add a dedicated subsection in related work, including **all the references you mentioned**. We also kindly emphasize that failure attribution is a fundamentally different research problem, and verifier, like PRMs are not directly applicable:
> PRMs are designed to reward well-structured reasoning chains in tasks like Math. In contrast, our failure attribution targets multi-agent systems, where turn sequences do not necessarily form a coherent reasoning chain. Single agent step may involve multiple reasoning steps—or none—while different agents may address distinct subtasks, with turn-taking reflecting task decomposition rather than unified reasoning. So, ProcessBench does not apply to our task.
>
> In response to your suggestions, we evaluated these PRMs in [3] for failure attribution under the same setting as Tab. 1 and shows the results in **https://shorturl.at/JSeJd**. It shows that no PRM outperforms All-at-Once and Step-by-Step in two metrics. This performance gap further underscores the fundamental difference between the objectives of PRMs and the proposed failure attribution task.
>
> **[2. Task formulation is subjective.]**
>
> We kindly clarify that the final annotation is not subjective. The disagreement number in Fig. 2 reflects only the **initial stage** of annotation, where approximately 20% cases were flagged as ambiguous. In subsequent stages, annotators rigorously followed the deterministic formulation in Sec. 2, engaging in discussion and cross-validation to ensure the final annotations were accurate.
>
> **[3. Why is the task challenging?]**
>
> It is challenging because:
> Agentic logs are very difficult to interpret. On average, each failure case includes 30 steps and 4,695 words and links to a total of 48 tools. Annotators must manually reconstruct agent actions, which requires understanding the complex action logic and manual run tool to verify behavior.
>
> Annotators may need to manually solve the task—either to obtain a "golden trajectory" for reference or to resume from intermediate states to assess whether a step constitutes a decisive error (as defined in Sec. 2), which demands replicating the agent’s conditions over multiple steps and is often hard.
>
> **[4. The results suggest the data do not contain self-correction. & How often is the decisive error step the same as the earliest error?]**
>
> Self-corrections are common in Who&When, which is reflected in the low average step-level accuracy of 14.2% in Tab. 1. The decisive-error step might occur after the prediction step of Step-by-Step.  In light of your suggestions, we conducted additional round of annotations—using the same procedure—solely on the hand-crafted systems (considering limited time) to identify the first-error step. We then compared it to the decisive error step and reported the overlap percentage in **https://shorturl.at/JSeJd**, which shows that in nearly half of the cases, they do not overlap.
>
> **[5. Dataset is small.]**
>
> We plan to expand the dataset in extended work. However, we respectfully argue that the current dataset is non-trivial and sufficient for drawing meaningful conclusions. Our main findings—such as the relative ranking of three methods across metrics—remain robust regardless of dataset scale. In Tab. 1, these rankings hold across both algorithm-generated and hand-crafted systems. The dataset includes 201 distinct agents, 4,092 steps, an average of 4,695 words per trajectory, and 48 external tools. Annotation required 84.3 hours by three experts. which offers a rich testbed for evaluating failure attribution methods.
>
> **[6. Were the results averaged over multiple runs?]**
>
> We did not initially run multiple LLM trials because: (1) the randomness has minimal impact given the task’s difficulty, (2) small variations don’t affect the paper’s core conclusions—such as the relative ranking of methods- and (3) extreme cost. In response to your suggestion, we conducted five more runs under the same setting as Tab. 4 and report the results in **https://shorturl.at/JSeJd**. We found the findings remain consistent.
>
> **[7.  Are the disagreements within a range of steps?]**
>
> No, the ambiguous decisive error between annotators does not exhibit a correlation with their positional distance.
>
> **[8. Regarding Fig. 4 and 5]**
>
> We will replace Fig. 5 with a table to improve clarity. Additionally, we will clarify the step ranges in Figure 4: Level 1 spans 5–17 steps, Level 2 covers 19–29, Level 3 includes 31–49, Level 4 ranges from 51–91, and Level 5 spans 93–130 steps.
>
> **Thank you for your time and consideration! We sincerely hope that you find our responses convincing and would consider increasing the rating.**

---

### Decision · Program_Chairs · 2025-05-01

**Decision:**

Accept (spotlight poster)

**Comment:**

This paper introduces the novel problem of automated failure attribution in LLM multi-agent systems, proposing the Who&When dataset and evaluating three methods to identify both the responsible agent and the decisive error step in failed trajectories.


The paper's strengths include:

- Novelty and importance of the problem formulation: The work introduces a well-motivated research area with strong potential impact in debugging LLM multi-agent systems (Reviewer micc, Reviewer Fdny, Reviewer BgjX).

- Introduction of a high-quality benchmark dataset: The Who&When dataset is carefully designed and annotated, providing a rigorous testbed for future research (Reviewer micc, Reviewer Fdny).

- Thorough experimental evaluation and clear empirical findings: The authors present comprehensive results with multiple methods, metrics, and ablations, yielding practical insights such as the influence of context length and model architecture (Reviewer micc, Reviewer Fdny, Reviewer BgjX).

- Strong writing and structured presentation: The paper is well-organized and accessible, with clear problem definitions, detailed methodological explanation, and thoughtful discussion (Reviewer micc, Reviewer Fdny).

Before the rebuttal, common concerns raised by reviewers included the limited dataset diversity and small size (Reviewer BgjX, Reviewer micc), subjectivity and potential ambiguity in the annotation process (Reviewer BgjX, Reviewer micc), and missing connections to related verifier-based work such as PRMs and Agent-as-Judge (Reviewer BgjX). These concerns were directly addressed in the rebuttal with additional experiments, clarification of annotation protocols, justification of task distinctions, and detailed discussion of scope, all of which were acknowledged by reviewers as sufficient.

Therefore, AC recommend acceptance of this paper.